# Fundamental Limits in Dissipative Processes during Computation

**DOI:** 10.3390/e21090822

**Published:** 2019-08-23

**Authors:** Davide Chiucchiú, Maria Cristina Diamantini, Miquel López-Suárez, Igor Neri, Luca Gammaitoni

**Affiliations:** 1Biological Complexity Unit, Okinawa Institute of Science and Technology and Graduate University, Onna, Okinawa 904-0495, Japan; 2NiPS Laboratory, Dipartimento di Fisica e Geologia, Universitá di Perugia, I-06100 Perugia, Italy; 3INFN Perugia, via A. Pascoli, I-06100 Perugia, Italy; 4ICMAB-CSIC, 08193 Barcelona, Spain

**Keywords:** dissipation, computing, fluctuations, heat, energy

## Abstract

An increasing amount of electric energy is consumed by computers as they progress in function and capabilities. All of it is dissipated in heat during the computing and communicating operations and we reached the point that further developments are hindered by the unbearable amount of heat produced. In this paper, we briefly review the fundamental limits in energy dissipation, as imposed by the laws of physics, with specific reference to computing and memory storage activities. Different from previous approaches, we will focus on the sole dynamics of the binary switches, the building blocks of the logic gates and digital memories, without invoking any direct connection to the notion of information.

## 1. Introduction

In the last fifty years, the Information and Communication Technology (ICT) sector has experienced a huge growth, mainly fostered by the ability of the underlying semiconductor industry to repeatedly scale down the size of the CMOS-FET (Complementary Metal-Oxide Semiconductor-Field Effect Transistor) switches, the building block of present computing devices, and to increase computing capability density up to a point where billions of switches have been assembled in a square centimetre. Further progress in this direction is thwarted by the amount of energy dissipated during switch operations: *the resulting power density for these switches at maximum packing density would be on the order of 1 ${\mathrm{MW}/\mathrm{cm}}^{2}$, orders of magnitude higher than the practical air-cooling limit.* [1]

There is little doubt that managing efficiently the use of energy, i.e., drastically reducing heat production, is a key aspect to be considered in designing future computing systems, especially for applications in smart sensors and Internet of Things devices, where the small dimension and the mobility characteristics require innovative solutions [2].

The energetic issues of future computers require a clear understanding of their functioning in terms of efficiency, i.e., the amount of operations performed per second, versus the amount of energy dissipated [3]. The search for today’s ultimate efficiency in computing is somehow similar to the search for the maximum efficiency in the functioning of heat engines that gave birth to the thermodynamics in the 18th and 19th centuries. However, different from the work done on steam engines, aimed at reaching the ultimate limit set by the Carnot theorem, here the ultimate physical limits in energy dissipation during computation is still a topic of debate [4]. The controversy is mainly associated with the role to be assigned to the notion of information, as introduced by Shannon [5] in the early 1940s, and applied by Ralph Landauer [6] in the framework of what is nowadays called the *Thermodynamics of information* [7]. Different from other approaches, in this paper, we will show that the notion of information, undoubtedly very useful in dealing with the mathematical aspects of computing tasks, is not necessarily required when one is interested in the mere dynamics of the machinery of the computing itself. As we will show in the following, we can carry-on our analysis of the energetic efficiency, just focusing on the functioning of the very basic physical elements of the digital computer: the *binary switches*. We show that the energetic analysis of the two main tasks of the digital computing process, i.e., logic-arithmetic operations and memory storage, can be performed without any recourse to the notion of information.

## 2. Binary Switches

Automatic digital computing is presently performed by manipulating binary logic states encoded in physical devices. The most relevant devices employed in such a task are logic gates and memories. In binary logic, where the logic states are just two, often identified with logic state 0 and logic state 1, the state manipulation is performed according to the Boolean logic and arithmetic operations are realised by the repeated application of basic logic operators called *logic gates*. All of the logic and arithmetic operations, and thus all digital computing, can be performed by assembling together sets of *universal* logic gates, like the NAND gate.

The NAND gate, whose logic output is 1 when and only when its two logic inputs are 0, and 0 otherwise, can be realised by employing two subsequent binary switches, as illustrated in Figure 1. The role of binary switch can be played by any practical device capable of assuming two different physical states (often denominated *open* and *close*) as a result of the application of an external force.

In order to perform logic and arithmetic operations, we need to change the state of a binary switch. In general, this can be realised by the application of a generalised force *F* that, by acting from outside on the switch, induces state changes, i.e., a *switch event*. We can call $F_{01}$$\left(F_{10}\right)$ the generalised force acting to bring the switch from 0 to 1 (1 to 0). Following this approach, we can easily identify two different classes of devices that can be usefully employed as binary switches. We will call them *combinational* and *sequential* switches.

In a *combinational* switch, we observe the following behaviour: when no external force is present, under equilibrium conditions, the switch is found in the logic state 0. When an external force $F_{01}$ is applied, it switches to the logic state 1 and remains in that state as long as the force is applied. Once the force is removed, it goes back to the logic state 0. A common example of a combinational switch is the electro-mechanical relay, used in many circuits involving lighting. Here, when a magnetic induction force is applied, the relay changes its state from open to closed and goes back to the initial state (open) when the force is removed. Another important example of a combinational switch is the transistor, widely employed in modern computers where the input force is represented by an externally applied voltage that makes the transistor switch from the high-impedance (non conducting) to a low-impedance (conducting) condition.

On the other hand, in a *sequential* switch, we observe a different behaviour: if the switch is in the logic state 0, it can be changed into the state 1 by applying an external force $F_{01}$, as before. However, in this case, when the force is removed, the switch remains in the logic state 1. This is true also for the switch event from 1 to 0 where the force $F_{10}$ has to be applied. We can say that, different from the combinational switch, the sequential switch remembers its state after the removal of the force. For such reason, they are good candidates for realising storage devices and are widely employed in computer memories.

### 2.1. Energy Dissipation in Charge-Based Switch Devices

As we have mentioned earlier, present computers are built using CMOS (*Complementary Metal-Oxide Semiconductor*) transistors, employed both as combinational and, in conjunction with capacitors, as sequential switches. For these devices, we face a number of dissipative effects that are due to their peculiar functioning and/or to the technology employed (semiconductor based, electric charge devices). For them, the most relevant source of dissipation during the switching [3] is associated with the charging and discharging of the electrical capacitances that are used to set the required voltage for the functioning of the transistor. If *C* is the capacitance involved, the amount of energy dissipated per switch event is twice the energy stored in the capacitor and amounts to:(1)$$E_{C}=CV^{2},$$ where *V* is the voltage across the capacitor plates. If the frequency of switching is *f*, we have a resulting dissipated power equal to
(2)$$P_{C}=\alpha CV^{2}f,$$ where α is a coefficient that ranges from 0 (extremely smooth switching) to 1 (square wave switching). This quantity is usually referred to as dynamic dissipation.

An additional source of dissipation arises from the so-called *static leakage* due to the presence of a subthreshold leakage phenomenon, i.e., small electric currents that occur between the source and drain of a transistor when it is in the subthreshold region and no conduction is expected. With the progressive reduction of the voltage *V*, operated with the aim of decreasing the dynamic dissipation due to the switching, the subthreshold leakage has increased its importance and with voltages as low as 0.2V, leakage can exceed 50% of total power consumption [8]. In absolute values, the amount of energy dissipated during a single switch event has been continuously reduced over the last forty years and has presently reached the value of approximately $10^{-17}$ J [3].

From the present description of the dissipation sources in charge-based switch devices, it appears clear that such phenomena are necessarily associated with the physical nature of the devices employed as binary switches. Perhaps it should be expected that, once we change the physical device, for example substituting the electronic transistor with a nano mechanical cantilever, the dissipative mechanisms change as well. Thus, in order to inspect the fundamental limits in energy dissipation for computing devices, we should proceed with identifying a general physical model of the binary switch and look for some dissipative mechanism that does not depend on the physical realisation of the switch itself and on the functioning principles being mechanical, optical, electro-mechanical or purely electronic.

### 2.2. The Physics of Binary Switches

Let’s consider the general description of a binary switch, as previously introduced. This can be represented in terms of a one degree-of-freedom dynamic system, subjected to a confining potential energy and external forces. If we assume x(t) as the relevant variable, we can define two identifiable states, e.g., $x<x_{TH}$ (logic state 0), $x>x_{TH}$ (logic state 1), where $x_{TH}$ is an arbitrary threshold value. The variable x(t) can represent an electric voltage (as in a transistor) or a position (as in a mechanical cantilever) or a value of the magnetisation (as in magnetic memories), just to mention a few relevant examples. Here, we are interested in those features of the variable x(t) that are general enough to be common to all the cited cases. The time evolution of x(t) can be described in terms of a second order differential equation, Langevin type:(3)$$\ddot{x}=-\frac{dU(x)}{dx}-\gamma \dot{x}+F\left(t\right)+\sigma \xi \left(t\right),$$ where U(x) is the confining potential, F(t) is the generalised force that produces the switch event, γ is the dissipative constant, assumed to be a viscous damping-like friction, and ξ(t) is a stochastic function with Gaussian distribution, zero mean and unitary standard deviation. The presence of a non negligible stochastic force is required due to the fact that present binary switches have reached small physical dimensions at the point that the presence of fluctuations cannot be neglected. At thermal equilibrium and in the presence of thermal noise as a dominant noise source, the spectral properties of ξ(t) are set by the corresponding fluctuation-dissipation relation.

The dynamics represented in (Equation 3) can accommodate both the *combinational* and the *sequential* switches, depending on the specific choice of the potential function U(x). Without loss of generality, here we will consider
(4)$$U\left(x\right)=a\frac{1}{2}x^{2}$$ for the combinational switch and
(5)$$U\left(x\right)=-a\frac{1}{2}x^{2}+b\frac{1}{4}x^{4}$$ for the sequential switch—with *a* and *b* being properly chosen constant parameters.

Due to the presence of the stochastic force, the system dynamics can be conveniently represented through the time evolution of the corresponding probability density p(x,t) that can be statistically computed from (Equation 3) or obtained as the solution of the associated Fokker–Planck equation [9]. The two states, 0 and 1, are realised with probability, respectively, with $p_{0}$ and $p_{1}$ ($p_{0}+p_{1}=1$) given by:(6)$$p_{0}=\int_{-\infty }^{x_{TH}}p(x,t)dx,\;p_{1}=\int_{x_{TH}}^{+\infty }p(x,t)dx,$$ with $x_{TH}$ a properly chosen threshold value. In Figure 2, we illustrate the two potentials U(x) with the associated probability densities p(x,t) corresponding to the logic states 0 (left) and 1 (right).

We note that, due to the presence of the fluctuating force, the stochastic process x(t) will oscillate around the minima of the potential U(x). For the sequential potential in (Equation 5), this implies that x(t) performs occasional random crossings between the two wells and, at equilibrium, with a symmetric potential and zero average fluctuating force, it requires that $p_{0}=p_{1}$.

## 3. Fundamental Energy Limits in Binary Switches

We are now interested in understanding what the minimum energy required is for operating binary switches.

In order to fix our ideas, let’s consider the switch dynamics defined by (Equation 3). For what we have seen so far, the logic switch event consists of the transformation of the probability density, such that, for the switch 0 to 1, the initial $p_{0}≃1$ and $p_{1}≃0$ become $p_{0}≃0$ and $p_{1}≃1$, while, vice versa, for the switch 1 to 0, the initial $p_{0}≃0$ and $p_{1}≃1$ become $p_{0}≃1$ and $p_{1}≃0$. It is clear that, in order to understand what the minimum energy involved is, we have to take into account all the possible energy processes associated with such a transformation [10]. Thus, in addition to taking into account the potential energy change and the role of friction, we also have to consider the change in the system entropy. Here, the system entropy associated with the stochastic process x(t), can be computed according to Gibbs as:(7)$$S=-k_{B}\int_{-\infty }^{+\infty }p\left(x,t\right)\ln p\left(x,t\right),$$ with $k_{B}$ being the Boltzmann constant. In order to understand what the minimum energy involved is, we will consider the two switch families separately.

### 3.1. Combinational Switches and Logic Gates

As we have anticipated above, combinational switches are the building blocks of logic gates that are realised by assembling together one or more combinational switches. Specifically, we need two switches to make a NAND gate that, being a universal gate, can be used in all the logic and arithmetic operations. Accordingly, the minimum energy required to operate a logic gate is identified once we define what the minimum energy required is to operate the combinational switch.

We note that logic states 0 and 1 do not have the same energy due to the shape of the potential in (Equation 4). If we indicate with $\Delta E=E_{1}-E_{0}$ the potential energy difference, it is clear that, in order to perform the switch, the deterministic force F(x,t) has to perform a work at least equal to ΔE, against the potential energy. What about the other forces? The stochastic force ξ(t) has zero mean; thus, on average, the work performed is zero. Finally, the dissipative force $\gamma \dot{x}$ performs a dissipative work that is proportional to the speed of the switch. In the adiabatic limit, i.e., when the switch event is produced with $\dot{x}\to 0$, the dissipated energy tends to zero. Due to the presence of the fluctuations, we have to consider the existence of a thermal bath that exchanges a certain amount of heat equal to TΔS. However, due to the linearity of the potential, it is always possible to choose a F(x,t) such that the system entropy at the beginning and at the end of the switch is the same. Thus, we have ΔS=0 and no minimum net contribution from heat is required. Based on this analysis, it is clear that the work done by the deterministic force F(x,t) during the 0 towards 1 switch is the opposite of the work done during the 1 towards 0 switch.

In summary, if we are capable of performing the switch event with arbitrarily small velocity, the amount of work done during the 0 to 1 switch being equal to the 1 to 0 switch, and, these events being on average equally likely in a long computation task, we can conclude that it is possible, at least in principle, to perform such a task by spending zero energy. In order to better illustrate this point, we performed an experiment whose results have been discussed in Ref. [11], where a micro electro-mechanical cantilever has been employed as a combinational switch. In Figure 3, we show the amount of energy dissipated during a cycle of two subsequent switches 0 to 1 to 0, performed at different velocities. It is apparent that, increasing the protocol time, the produced heat decreases following a power law ($\tau_{p}^{-3}$), in good agreement with a friction model used to account for the velocity dependent dissipation [11].

### 3.2. Sequential Switches and Memory Devices

If combinational switches are the building blocks of logic gates, sequential switches are the basic components of digital memories.

In this case, two different operations have to be considered in order to account for the energy dissipation during their functioning. The first operation is called *reset* and takes place when the sequential switch is at equilibrium. In this case, in order to write a memory bit, we need to break the equilibrium condition and to re-set the binary switch in one of the two logic states. The second operation is the *switch* and it is carried out when the initial state is known and one needs to change it.

#### 3.2.1. The Reset Operation

As we have previously mentioned, the equilibrium condition of x(t) according to (Equation 3) and (Equation 5) is represented by a symmetric probability density p(x,t), where $p_{0}=p_{1}=0.5$ (see Figure 4, left). In order to store a binary digit, we need to set our memory in a given state, 0 or 1. This operation requires a change in the probability density that becomes entirely confined within one well of the potential. In Figure 4, we show the change in p(x,t) for the *reset to 0* operation. Here, we have to deal with the heat exchanged with the thermal bath because the system entropy at the beginning and at the end of the switch is not the same. It is easy to show that the required change in entropy is $\Delta S=-k_{B}\log2$ and thus a minimum net contribution of $Q=k_{B}T\log2$ is required. Such a work is realised by the deterministic force. Finally, the dissipative work done by the frictional force can be reduced to zero if a proper adiabatic protocol is followed.

#### 3.2.2. The Switch Operation

The switch event is pictorially represented in the right column of Figure 2, where p(x,t) associated with the logic state 0 is shown in the upper graph. Here, $p_{0}≃1$ and $p_{1}≃0$. The switch event consists of changing the p(x,t) into that one represented in the lower graph, associated with the logic state 1 where $p_{0}≃0$ and $p_{1}≃1$. Different from the previous case of the combinational switch, here, the logic states 0 and 1 have the same energy thanks to the potential symmetry in Equation (Equation 5) and no net work is required by the deterministic force F(x,t). Once again, particular care has to be devoted in selecting a switch protocol that keeps the system as close to equilibrium as possible (adiabatic transformation) in order to minimise the action of the dissipative force $\gamma \dot{x}$. We observe that this requirement can be difficult to satisfy. As a matter of fact, there are two conflicting requirements: on one side, we want to perform a switch that is slow enough to dissipate little or no energy at all and, on the other hand, we want to perform the switch in a time that is shorter than the relaxation time that brings the p(x,t) to its relaxed state with $p_{0}=p_{1}=0.5$. In the paper by Gammaitoni et al. [12], a possible protocol capable of satisfying such a requirement is illustrated together with some experiment performed by using nano magnets. In summary, also in this case, the minimum heat exchanged with the thermal bath can be zero, by the moment that the change in entropy is zero.

#### 3.2.3. Memory Preservation

Fundamental for the functioning of the memory is that the potential barrier allows for statistically confining x(t) for a given time within one of the two wells (Figure 2, right), hence ensuring that one given bit is effectively stored. As we have anticipated, this confined state represents a non-equilibrium condition that evolves, within the system relaxation time $\tau_{k}$, to statistical equilibrium (Figure 5). This process is described via the time evolution of the probability density function p(x,t) as follows. Let us assume we have a memory where bit 1 is stored. The initial probability density p(x,0) shows a sharp peak centred in the right well (Figure 5, leftmost panel). According to the dynamics of the system, p(x,t) will first relax inside the right well and then it will diffuse into the left well, thus developing a second peak. At any given time *t*, the probability that the system encodes the wrong logic state is represented by $p_{0}\left(t\right)$ that grows towards the equilibrium condition $p_{0}=0.5$ when the memory is statistically lost (Figure 5, rightmost panel).

In order to avoid memory loss, a periodic refresh procedure is performed on any binary switch. This procedure consists of reading and then writing back the content of the memory, and it is performed at intervals $t_{R}$ [13]. The refresh operation restores a non-equilibrium condition by shrinking the width of each peak of p(x,t) back to its original state and requires some energy to be dissipated. This is due to the fact that, during this transformation, external work is done by the deterministic force in order to reduce the entropy associated with the increased width of the distribution. Such a work can be estimated by measuring the entropy change. In addition, we have to consider the losses associated with the friction if the transformation speed is not kept small enough.

The entropy change can be computed [14] by assuming that the dynamics of x(t) is confined within one well and it can be approximately described by the dynamics of a harmonic oscillator, characterised by a Gaussian probability density function. Such an approximation is valid if $t_{R}$ is much smaller that the global relaxation time $\tau_{k}$. In this perspective, the change in entropy can be computed as:(8)$$\Delta S=k_{B}\ln\left(\frac{\sigma_{i}}{\sigma_{f}}\right),$$ where $\sigma_{i}$ is the target standard deviation of the Gaussian peak to be achieved with the refresh and $\sigma_{f}$ is the standard deviation of the Gaussian peak before the refresh. While $\sigma_{i}$ can be arbitrarily chosen, $\sigma_{f}$ depends on $t_{R}$ [14]. Hence, the minimum required energy to operate a single refresh is Q=TΔS.

In order to test the practical attainability of this result, we performed an experiment [14] by employing a micro electro-mechanical cantilever. A tiny NdFeB magnet is attached to the cantilever tip and an external electromagnet is placed in front of the cantilever in order to change the potential stiffness by changing the distance between the two magnets. The probability density change associated with the refresh operation is obtained by changing the stiffness of the potential.

In Figure 6, we show the measured values of *Q* required to perform a single refresh operation as a function of the protocol time $t_{p}$, for fixed $\sigma_{i}$ and $\sigma_{f}$. We can see that *Q* approaches the minimum value given by Equation (Equation 8) when $t_{p}$ increases towards the quasi-static protocol condition. In a practical memory, such a condition is clearly attainable only if the required protocol time satisfies the given condition $t_{p}\ll t_{R}\ll \tau_{k}$.

Provided that we want to keep the memory, i.e., preserving the stored bit with a probability of memory failure smaller than a given $P_{E}$, for a general time $\bar{t}$, we computed [14] the minimum energy dissipation required as:(9)$$Q_{m}=-NT\Delta S=\frac{\bar{t}}{t_{R}}k_{B}T\ln\left(\frac{\sqrt{\sigma_{w}^{2}+e^{-\frac{t_{R}}{\tau_{w}}}\left(\sigma_{i}^{2}-\sigma_{w}^{2}\right)}}{\sigma_{i}}\right),$$ where *N* is the number of subsequent refresh operations performed, $\sigma_{w}$ and $\tau_{w}$ are, respectively, the equilibrium standard deviation and the relaxation time inside one well. As it is well apparent, such an amount of energy dissipated can be, in principle, reduced to zero, provided that the probability distribution is kept constantly close to the equilibrium distribution and/or if the refresh time $t_{R}$ is arbitrarily small.

## 4. Conclusions

In conclusion, we have briefly reviewed the fundamental limits in energy dissipation, as imposed by the law of physics, when basic computing tasks are performed. We have shown that logic gate operations, made by operating sets of combinational switches, can be, at least in principle, performed without any spending of energy, provided an optimal protocol is followed [15,16,17,18]. The same conclusion can be applied when sequential switches are operated as one-bit memory devices. The only exception is represented by the *reset* operation, necessarily required when a memory device is written for the first time. We have also shown that, once a one-bit memory has been written, its content can be kept for a given (finite) time $\bar{t}$ without spending any energy, provided the refresh operation is performed often enough so that the system does not change its entropy significantly.

Although the focus of this work is on *fundamental limits*, one might wonder what help it could be to know that, in principle, a computer can be operated by spending zero energy but, in practice, this is obtained only under the condition that all the operations are performed adiabatically, i.e., extremely slowly. In our opinion, the zero-energy limit, once established without any controversy, is a prospective goal that can lead the future research for novel, more energy-efficient technologies, compared to the present CMOS. In order to fix the ideas, we notice that the most performant present commercial switches take approximately $10^{-12}$ s to switch with a dissipated power of $10^{-4}$ W. In comparison, a micro-mechanical logic switch, built according to [11] would take approximately $10^{-3}$ s to switch with a dissipated power of $10^{-19}$ W. For an analogy, a biological neuron takes approximately $10^{-2}$ s to switch with a dissipated power of $10^{-14}$ W [19].

## Figures and Tables

**Figure 1 entropy-21-00822-f001:**
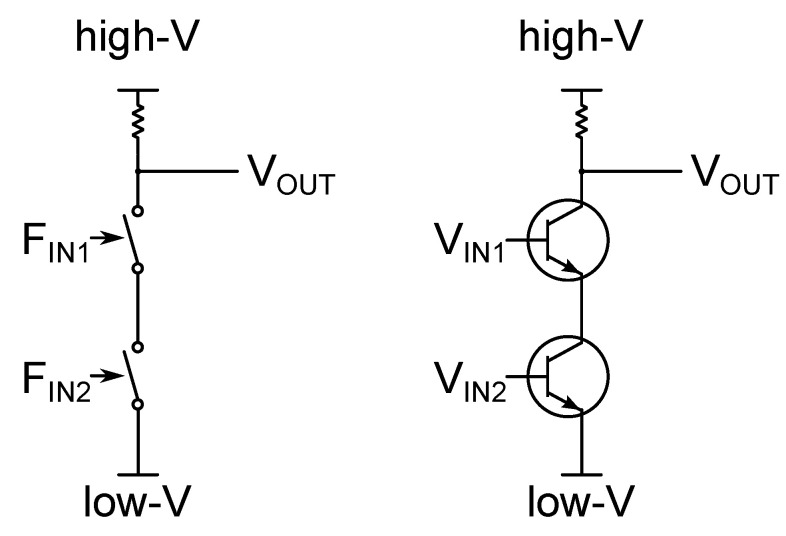
Universal logic gate NAND. (**Left**) The logic output OUT is here associated with an electric voltage value across a simple circuit: high-V corresponds to the logic state 1 and low-V corresponds to the logic state 0. Binary switches are represented here as mechanical switches that can assume the logic state “0” (physical state *open*) or the logic state “1” (physical state *close*). When both the switches are in the close position, the circuit behaves as a simple conductor and the voltage $V_{OUT}$ position assumes the value low-V. (**Right**) Logic gate NAND implemented using transistors as binary switches. In this case, both the input logic state and the output logic state are physically encoded into electrical voltages.

**Figure 2 entropy-21-00822-f002:**
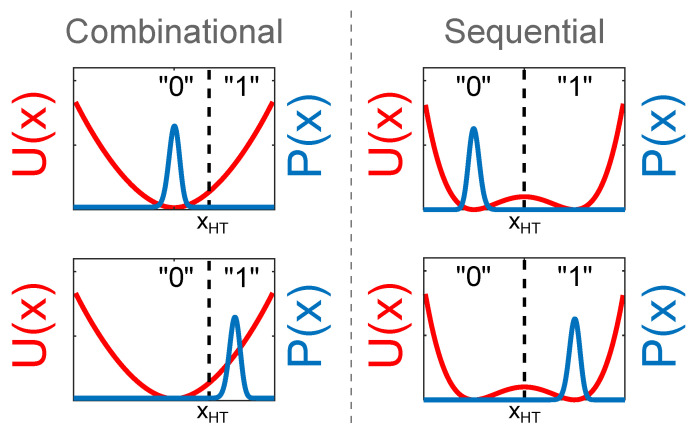
Potential functions U(x) with the associated probability densities p(x,t). (**Left column**) potential function U(x) for the combinational switch. Here, the threshold $x_{TH}=1.2$. Upper graph: p(x,t) associated with the logic state 0. Here, $p_{0}≃1$ and $p_{1}≃0$. Lower graph: p(x,t) associated with the logic state 1. Here, $p_{0}≃0$ and $p_{1}≃1$. (**Right column**) potential function U(x) for the sequential switch. Here, the threshold $x_{TH}=0$. Upper graph: p(x,t) associated with the logic state 0. Here, $p_{0}≃1$ and $p_{1}≃0$. Lower graph: p(x,t) associated with the logic state 1. Here, $p_{0}≃0$ and $p_{1}≃1$.

**Figure 3 entropy-21-00822-f003:**
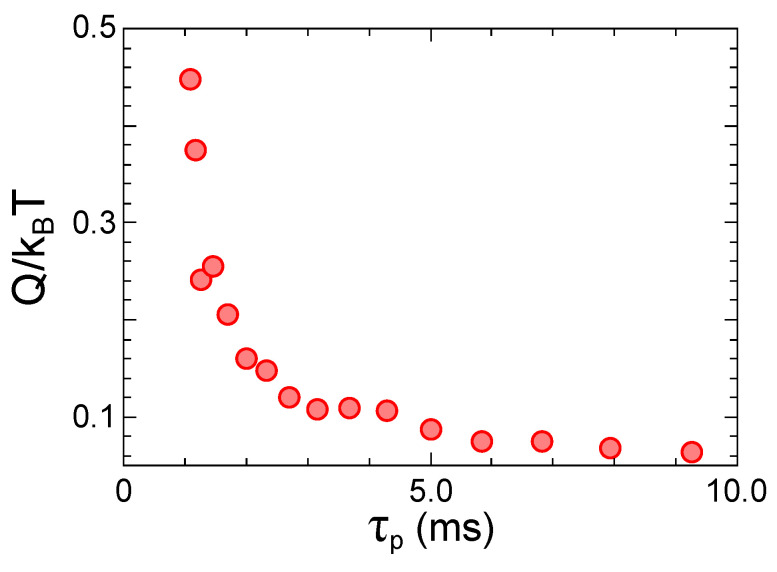
Average heat production during the cycled combinational switch operation as a function of protocol time $\tau_{p}$. By increasing the protocol time, the produced heat decreases following a power law. The average heat *Q* is shown here in $k_{B}T$ units, where $k_{B}$ is the Boltzmann constant and *T* is the room temperature.

**Figure 4 entropy-21-00822-f004:**
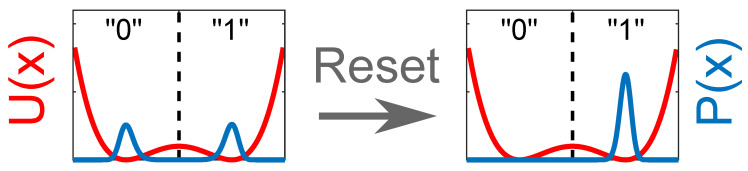
*Reset to 0* operation. (**Left**) probability density p(x,t) at equilibrium. In this condition, $p_{0}=p_{1}=0.5$. (**Right**) probability density p(x,t) after the reset operation. In this condition, $p_{0}≃1$ and $p_{1}≃0$. During the reset operation, the system entropy decreases by an amount $\Delta S=-k_{B}\log2$. The same amount of entropy change is required for the *reset to 1* operation.

**Figure 5 entropy-21-00822-f005:**
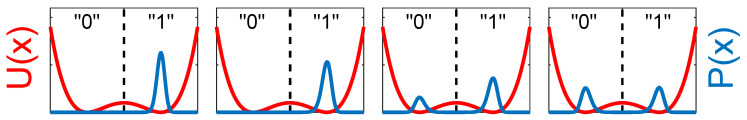
Panels from left to right show the memory-loss mechanism when the bit 1 is initially stored. The curves give a qualitative time evolution of p(x,t) as the relaxation to equilibrium process takes place.

**Figure 6 entropy-21-00822-f006:**
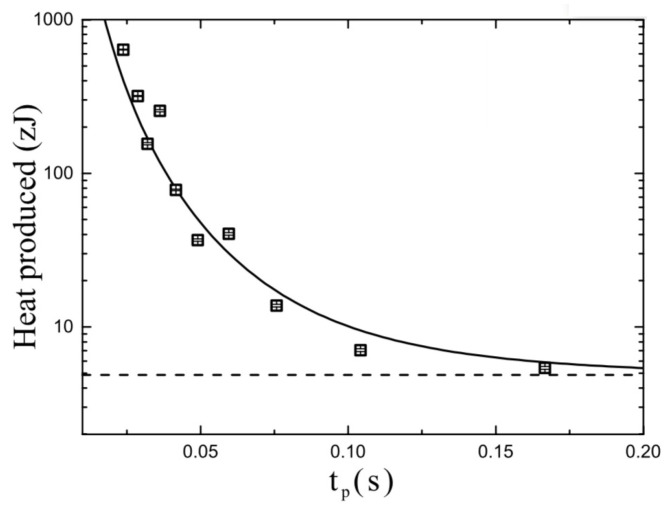
Experimental results of produced heat for a single refresh operation as a function of the protocol time $t_{p}$. By increasing $t_{p}$, the produced heat tends to the lower bound Q=−TΔS represented here by the dashed line. The solid squares represent the heat from the experiment, while the solid line is the fit with the Zener dissipative model.

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
