# Peer review of "Fundamental Limits in Dissipative Processes during Computation"

_entropy, 2019, doi:10.3390/e21090822_

Round 1
Reviewer 1 Report
In this manuscript the authors review the limits of dissipation of
solid state binary switches in to different arrangements:
``combinational'' consisting in a harmonic confining potential and
``sequential'', consisting in a quartic potential. They approach
makes not use of the notion of information but exclusively on the
dynamics of the elements investigated.
The authors identify the dissipated energy in both cases by modelling
the evolution of the process as a stochastic Langevin process Eq.(3).
For the combinational switch they find that due to the linearity of
the potential the entropy change is zero, meaning that the work done
during the process is zero, so that it can be performed by spending
zero energy. For sequential switches, the authors show that while the
reset operation involves a dissipation of at least k_BT log 2
(Landauer principle), the switch operation dissipates an amount of
energy that decays with the operation time.
The results presented in this manuscript are interesting and sound,
and I recommend its publication in Entropy once the authors have
addressed the minor point below.
One minor question (Fig 3.): The authors state that in cycled
combinatorial switches the heat decays as a power law with the
duration of the protocol. Could the authors please indicate the power
scaling of it?
Author Response
We would like to thank the referee for his/her comments.
About his/hers observation:
"One minor question (Fig 3.): The authors state that in cycled
combinatorial switches the heat decays as a power law with the
duration of the protocol. Could the authors please indicate the power
scaling of it?"
We have specifically added the power scaling by modifying the sentence in line 177 as follows:
It is apparent that increasing the protocol time, the produced heat decreases following a power law ($\tau_p^{-3}$), in good agreement with…
Finally, we have spell checked the ms.
Reviewer 2 Report
This manuscript considers the limits of energy dissipation associated with the operations of a computer, using a statistical thermodynamical approach. In it, experimental results obtained by some of the authors are related to derivations of the minimal theoretical heat generation that can be achieved, showing that in the limit of a slow protocol, these minima can be approached quite closely.
The conclusion that a very slow protocol reduces energy dissipation is consistent with what we might expect from undergraduate thermodynamics. While I have no problem with the specific treatment developed here, I’m not sure that it adds much to our understanding beyond arguments one might make for example in analogy with the adiabatic compression of a piston (and beyond the already-published experimental data). However, the theory here is perhaps developed in closer analogy to the experiments (perhaps the authors could clarify if this is the motivation for the choices of potential and the arbitrary position of x_HT for the harmonic case), so I’m willing to recommend the manuscript be accepted despite the somewhat unsurprising nature of the results when all is said and done.
Perhaps of more interest is: how feasible is it to apply a slow protocol in a computer, whose primary purpose is usually to perform calculations as quickly as possible? What would the trade-off be between reducing dissipation and reducing calculation time, and is it feasible given the other timescales that must be negotiated (the relaxation and rewrite time scales)? Comments on these points would align the manuscript more closely with the sort of insights one might expect from its title.
I note that ‘cure’ should be ‘care’ on p7, and the usual phrase is ‘without loss of generality’.
Author Response
We would like to thank the referee for hers/his comments.
In the following we specifically address each one of them.
The referee wrote:
This manuscript considers the limits of energy dissipation associated with the operations of a computer, using a statistical thermodynamical approach. In it, experimental results obtained by some of the authors are related to derivations of the minimal theoretical heat generation that can be achieved, showing that in the limit of a slow protocol, these minima can be approached quite closely.
The conclusion that a very slow protocol reduces energy dissipation is consistent with what we might expect from undergraduate thermodynamics. While I have no problem with the specific treatment developed here, I’m not sure that it adds much to our understanding beyond arguments one might make for example in analogy with the adiabatic compression of a piston (and beyond the already-published experimental data).
Our answer:
We agree, indeed with these comments. To us, is all quite clear and straightforward. The fact that the minimum amount of energy dissipation from operating logic gates has represented a very controversial topic for more than five decades is a matter that will amuse future historians of physics (see e.g. Wolfgang Porod, The Thermodynamics of Computation: A Contradiction. In {\em Energy Limits in Computation}, 141-154, Springer, 2018).
The referee wrote:
However, the theory here is perhaps developed in closer analogy to the experiments (perhaps the authors could clarify if this is the motivation for the choices of potential and the arbitrary position of x_HT for the harmonic case), so I’m willing to recommend the manuscript be accepted despite the somewhat unsurprising nature of the results when all is said and done.
Our answer:
The choice of the harmonic potential has been made in analogy with the experiment and the principle of the simplest potential capable of capturing the required features. The value of the threshold x_HT depends on the experimental realization. In our case a micro cantilever with few nanometers displacement.
The referee wrote:
Perhaps of more interest is: how feasible is it to apply a slow protocol in a computer, whose primary purpose is usually to perform calculations as quickly as possible? What would the trade-off be between reducing dissipation and reducing calculation time, and is it feasible given the other timescales that must be negotiated (the relaxation and rewrite time scales)? Comments on these points would align the manuscript more closely with the sort of insights one might expect from its title.
Our answer:
The referee is right. We have added the following sentence:
“Although the focus of the ms is on fundamental limits, one might wonder what help it could be to know that in principle a computer can be operated by spending zero energy but in practice this is obtained only under the condition that all the operations are performed adiabatically, i.e. extremely slow. Well, in our opinion the zero-energy limit, once established without any controversy, is a perspective goal that can lead future research for novel, more energy-efficient technologies, compared to the present CMOS. In order to fix the ideas we notice that that most performant present commercial switches take approximately $10^{-12} s$ to switch with a dissipated power of $10^{-4} W$. In comparison a micro-mechanical logic switch built according to \cite{below} would take approximately $10^{-3} s$ to switch with a dissipated power of $10^{-19} W$. For an analogy, a biological neuron takes approximately $10^{-2} s$ to switch with a dissipated power of $10^{-14} W$ \cite{SRA}.”
The referee wrote:
I note that ‘cure’ should be ‘care’ on p7, and the usual phrase is ‘without loss of generality’.
Our answer:
We thank the referee. Both points have been corrected.
This manuscript is a resubmission of an earlier submission. The following is a list of the peer review reports and author responses from that submission.
Round 1
Reviewer 1 Report
There is an important correct statement of the paper: The information entropy change has nothing to do with the change of thermal entropy and heat in a logic device. But its sore and implication is not analyzed in depth and neither is the paper [Kish & Ferry, PLA 2018, attached] that does this job and gets to the same conclusion is cited.
The paper has incorrect conclusion that zero energy dissipation during a logic state change can be approached; particularly that the dissipation can be brought below the Szilard-Brillouin limit of kT ln (2) (often incorrectly called Landauer limit).
The main error on which this claim is based is Equation 2. It comes from earlier errors of Cavin and coworkers just like the error to reduce energy dissipation by refreshing the logic state (or memory).
The invalidity of Eq. 2 is obvious if we investigate the circuit for a totally smooth charging, called "current-generator charging" deeper. Indeed the energy dissipation on the implicitly assumed serial resistor (or the relevant loss term in Eq. 3 in other systems) goes to zero when the current is approached to zero, so seemingly zero energy dissipation can be reached. However a current generator, or any other charging mechanism that deviates from the exponential relaxation of an RC element (or overdamped other system), is an active elements and it can easily be shown the lower limit of the energy dissipation is exactly the same value as the energy we have seemingly gained back. It is the same conclusion for the alternative situation (not shown in this paper) when the voltage is stepped up in small steps.
In conclusion, the paper and its main conclusions are flawed. To fix these, the whole paper should be rewritten; the **correct** analysis could be to show an example why Landauer theory is wrong: No information entropy change is present.
Note: the manuscript lists a related Nature article (Ref. [9]) of some of the authors but does not list the critical comment that invalidates that article (Kish, FNL 2016; attached).
Finally: the "experiment" and some of the scheme mentioned are extremely poorly described. This is unfortunately typical nowadays. No circuitry, or scheme, no specific data about the elements, etc, thus they are impossible to reproduce in a counter-experiment.
